# Peripheral chemoreflex restrains skeletal muscle blood flow during exercise in participants with treated hypertension

Ana Luiza C. Sayegh[1] , Michael J. Plunkett[1] , Thalia Babbage[1], Mathew Dawes[2], Julian F.R. Paton[1] and James P. Fisher[1] 

[1]*Department of Physiology, Manaaki Manawa – The Centre for Heart Research, Faculty of Medical & Health Sciences, University of Auckland, Auckland, New Zealand*
[2]*Department of Medicine, Faculty of Medical & Health Sciences, University of Auckland, Auckland, New Zealand*

Handling Editors: Vaughan Macefield & Marc Kaufman

The peer review history is available in the Supporting Information section of this article (https://doi.org/10.1113/JP286998#support-information-section).

**Abstract** We tested the hypothesis that in human hypertension, an increased tonicity/sensitivity of the peripheral chemoreflex causes a sympathetically mediated restraint of nutritive blood flow to the exercising muscles. Fourteen patients with treated hypertension (age $69 \pm 11$ years, $136 \pm 12/80 \pm 11$ mmHg; mean $\pm$ SD) were studied under conditions of intravenous 0.9% saline (control) and low-dose dopamine ($2 \ \mu g \ kg^{-1} \ min^{-1}$) to inhibit the peripheral chemoreflex, at baseline, during isocapnic hypoxic rebreathing and during rhythmic handgrip exercise (3 min, 50% maximum voluntary contraction). At baseline, dopamine did not change mean

blood pressure (95 ± 10 *vs.* 98 ± 10 mmHg, $P = 0.155$) but increased brachial artery blood flow (59 ± 20 *vs.* 48 ± 16 ml min$^{-1}$, $P = 0.030$) and vascular conductance (0.565 ± 0.246 *vs.* 0.483 ± 0.160 ml min$^{-1}$ mmHg$^{-1}$; $P = 0.039$). Dopamine attenuated the increase in mean blood pressure (Δ3 ± 4 *vs.* Δ8 ± 6 mmHg, $P = 0.007$) to isocapnic hypoxic rebreathing and reduced peripheral chemoreflex sensitivity by 28 ± 37% ($P = 0.044$). Rhythmic handgrip exercise induced increases in brachial artery blood flow and vascular conductance (both $P < 0.05$ *vs.* rest after 45 s) that were greater with dopamine than saline (e.g. Δ76 ± 54 *vs.* Δ60 ± 43 ml min$^{-1}$ and Δ0.730 ± 0.440 *vs.* Δ0.570 ± 0.424 ml min$^{-1}$ mmHg$^{-1}$, respectively, at 60 s; main effect of condition both $P < 0.0001$). Our results indicate that the peripheral chemoreflex is tonically active at rest and restrains the blood flow and vascular conductance increases to exercise in treated human hypertension.

(Received 26 May 2024; accepted after revision 1 August 2024; first published online 14 September 2024)

**Corresponding author** James P. Fisher: Department of Physiology, Manaaki Manawa — The Centre for Heart Research, Faculty of Medical & Health Sciences, University of Auckland, 85 Park Road, Grafton, Auckland 1142, New Zealand. Email: jp.fisher@auckland.ac.nz

**Abstract figure legend** Cardiorespiratory and brachial artery haemodynamic assessments were made during intravenous infusion of low-dose dopamine, to inhibit the peripheral chemoreflex, and saline (control) at rest, and during hypoxia and rhythmic handgrip exercise in people with hypertension. Compared with saline, low-dose dopamine reduced resting ventilation, the hypoxic ventilatory response and increased the magnitude of the increase in brachial blood flow and vascular conductance during rhythmic handgrip.

## Key points

- It was hypothesised that in human hypertension, an increased tonicity/sensitivity of the peripheral chemoreflex causes a sympathetically mediated restraint of nutritive blood flow to the exercising muscles.
- Treated patients with hypertension ($n = 14$) were studied under conditions of intravenous 0.9% saline (control) and low-dose dopamine (2 µg kg$^{-1}$ min$^{-1}$) to inhibit the peripheral chemoreflex.
- Low-dose dopamine reduced resting ventilation and peripheral chemoreflex sensitivity, and while mean blood pressure was unchanged, brachial artery blood flow and vascular conductance were increased.
- Low-dose dopamine augmented the brachial artery blood flow and vascular conductance responses to rhythmic handgrip.
- These findings indicate that the peripheral chemoreflex is tonically active at rest and restrains the blood flow, and vascular conductance increases to exercise in treated human hypertension.

## Introduction

Hypertension is a major risk factor for coronary artery disease, stroke and heart failure (Elliott, 2007; Kjeldsen, 2018) and is, therefore, a global public health problem (Williams et al., 2018). Conversely, exercise training and regular engagement in physical activity can help lower blood pressure, thus reducing the risk of cardiovascular disease (Blair et al., 1996). However, exercise-induced surges in blood pressure are exaggerated in hypertension (Aoki et al., 1983; Sumimoto et al., 1991),

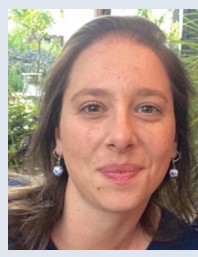

**Ana Luiza Sayegh** completed her PhD at the University of São Paulo, Brazil. She is currently a research fellow in the Department of Physiology at the University of Auckland. Ana specializes in integrative human physiology with a focus on autonomic control and the cardiovascular and respiratory systems. Her research explores the impact of various cardiopulmonary conditions, such as hypertension, heart failure and pulmonary arterial hypertension, on the neural regulation of circulation.

contributing to the incidence of stroke and myocardial infarction (Pescatello et al., 2004; Rothwell, 2010), but these exercise-induced surges in blood pressure are not reduced by current anti-hypertensive treatments (Chant et al., 2018).

During exercise, blood flow to the active skeletal muscles increases to provide the oxygen needed to meet the raised metabolic demands of the tissue (Hellsten et al., 2012; Joyner & Casey, 2015). Sympathetic vasoconstrictor outflow to inactive regions (e.g. splanchnic circulation) also increases during exercise, which facilitates the redistribution of cardiac output to the active muscles. To maintain blood pressure, raised sympathetic nerve activity competes with local vasodilator metabolites to restrain the hyperaemic response of the exercising muscles (Fisher et al., 2015). However, if this sympatho-excitation is excessive and the requirements of the active skeletal muscles for nutritive blood flow are not met, premature fatigue, exercise intolerance and exaggerated surges in blood pressure can result (Joyner & Casey, 2015).

Exercise-induced increases in sympathetic nerve activity are generally attributed to feedforward signals from higher brain centres (central command) and feedback signals from metabolically and mechanically sensitive groups III and IV skeletal muscle afferents (exercise pressor reflex) (Fisher et al., 2015). Although classically recognised as the body's major oxygen sensor and for reflexively increasing ventilation (Curran et al., 2000; Olson et al., 1988), the carotid chemoreceptors have also been demonstrated to drive, in part, the sympathetically mediated restraint of blood flow to the exercising skeletal muscles (Stickland et al., 2007, 2008). Moreover, peripheral chemoreflex sensitivity is increased during exercise (Forster et al., 1974), due at least in part to the activation of the exercise pressor reflex (Oliveira et al., 2024). Notably, in dogs with heart failure, the infusion of low-dose dopamine to transiently inhibit the carotid chemoreflex produced hindlimb vasodilatation during exercise, and this response was abolished following carotid body denervation (Stickland et al., 2007). As this vasodilator response was abolished with $\alpha$-adrenergic blockade, it was concluded that this was mediated by a reduction in sympathetic vasoconstrictor activity (Stickland et al., 2007).

Several studies, including our own, have identified raised peripheral chemoreflex tonicity and sensitivity (Abdala et al., 2012; McBryde et al., 2013; Pijacka et al., 2016) and elevated resting sympathetic nerve activity (Abboud, 1982; Esler, 2000; Grassi & Ram, 2016; Warnert et al., 2016) in animal models and humans with hypertension. There is also evidence to suggest that the exercise pressor reflex is augmented in hypertension (Barbosa et al., 2016; Delaney et al., 2010; Leal et al., 2013; Sausen et al., 2009), which may serve to further increase peripheral chemoreflex sensitivity during exercise (Oliveira

et al., 2024). However, it is currently unknown whether the aberrant afferent activity of the peripheral chemoreflex drives a sympathetically mediated restraint of blood flow to the exercising muscles in human hypertension. Therefore, in the present study, we tested the hypothesis that the infusion of low-dose dopamine to inhibit the peripheral chemoreflex (Edgell et al., 2015; Niewinski et al., 2014; Stickland et al., 2007, 2011) will augment blood flow to exercising skeletal muscle in participants with hypertension.

## Methods

### Ethical approval

The study was approved by the Northern B Health and Disability Ethics Committee, Auckland, New Zealand (19/NTB/125), registered in the Australian New Zealand Clinical Trials Registry (ACTRN12619001767190) and conducted in accordance with the *Declaration of Helsinki* (2013). All participants gave written consent after being provided with a detailed written and verbal explanation of the study procedures.

### Participants

Fourteen participants (mean $67 \pm 11$ years, range 39–83 years) with essential hypertension (Stage 2, treated controlled or uncontrolled, office systolic blood pressure (BP) $\geq 140$ mmHg or diastolic BP $\geq 90$ mmHg (Williams et al., 2018)) were recruited for the study. All participants reported being non-smokers, not users of recreational drugs, and not abusers of alcohol. All patients were treated with at least one anti-hypertensive medication, and no participant was taking prescribed medication aside from anti-hypertensive therapy. No women were users of hormone replacement therapy. Exclusion criteria involved: body mass index $<18$ and $>30$ kg/m$^2$, acute or chronic disorders associated with alterations in cardiovascular structure or function, and a history of pulmonary, metabolic, or neurological disease. Participant characteristics and medication usage are provided in Table 1.

### Experimental protocol

Participants attended two laboratory visits: a familiarisation visit and an experimental visit. At the familiarisation visit, anthropometric, demographic and clinical information were collected, and the participants were familiarised with all study measurements and procedures, aside from the intravenous infusion.

Participants were asked to refrain from taking their morning medications on the day of the study. These were

**Table 1. Participant characteristics**

| Demographics | |
|---|---|
| *N* | 14 |
| Age (years) | 67 ± 11 |
| Women, *n* (%) | 8 (57) |
| Weight (kg) | 70.1 ± 14.5 |
| Height (m) | 1.67 ± 0.09 |
| BMI (kg m$^{-2}$) | 25.1 ± 4.2 |
| Time since diagnosis (years) | 6 ± 7 |
| *Ethnicity* | |
| European, *n* (%) | 9 (65) |
| Māori, *n* (%) | 2 (14) |
| Pacific peoples, *n* (%) | 1 (7) |
| Asian, *n* (%) | 1 (7) |
| Other, *n* (%) | 1 (7) |
| *Medication used* | |
| ACE inhibitor, *n* (%) | 4 (29) |
| ARBs, *n* (%) | 3 (21) |
| $\beta$ blocker, *n* (%) | 1 (7) |
| Ca$^{2+}$ channel blocker, *n* (%) | 4 (29) |
| Statin, *n* (%) | 5 (36) |
| Thiazide diuretic, *n* (%) | 3 (21) |

Values are expressed as the mean ± SD for normally distributed data and as frequency (percentage) for discrete variables. Abbreviations: ACE, angiotensin-converting enzyme; ARBs, angiotensin receptor blockers; $\beta$, beta; BMI, body mass index; Ca$^{2+}$, calcium.

then taken after the protocol before leaving the laboratory. At the experimental visit, participants were asked to lie in a semi-recumbent position on a bed and a small-bore (23 gauge) intravenous catheter was inserted in the antecubital fossa/hand by a clinically qualified investigator (M.J.P.). Participants were then instrumented for measurement of heart rate, BP, ventilation, arterial oxygen saturation and brachial artery blood flow. Two trials were performed according to a single-blind randomised cross-over design. Dopamine was administered intravenously via an intra-venous catheter at a continuous infusion of 2 µg kg min$^{-1}$ (dopamine trial) as previously described (Edgell et al., 2015; Niewinski et al., 2014; Stickland et al., 2011). In the control trial, normal saline (0.9% sodium chloride solution) was continuously infused at an equal volume rate as in the dopamine trial. Following saline and dopamine infusion, a 10 min wash-in period was taken (Boetger & Ward, 1986). Each trial consisted of: (1) 5 min resting baseline, (2) isocapnic hypoxia rebreathing (Fan et al., 2011), (3) 15 min recovery until restoration to baseline of measured variables, (4) rhythmic handgrip at 50% maximal voluntary contraction for 3 min (1 s contraction, 2 s relaxation) (Stickland et al., 2008). At the end of rhythmic handgrip exercise, the infusion was stopped and a 30 min wash-out period was conducted (i.e. well

in excess of the half-life for dopamine in the systemic circulation which is 1–5 min) (Sonne et al., 2024).

For the isocapnic hypoxia rebreathing protocol, participants breathed through a mouthpiece and, after the resting baseline, at the end of a normal expiration, were connected to a closed circuit filled with room air (21% $O_2$, balance $N_2$) by a three-way stopcock. Rebreathing elicited a progressive decrease in $PO_2$, while isocapnia was maintained by the manual regulation of airflow via a soda lime scrubber using a second three-way stopcock (Fan et al., 2011). Isocapnic hypoxia rebreathing was terminated either when the partial pressure of end-tidal oxygen ($P_{ET}O_2$) reached 45 mmHg or at the participant's discretion. After each trial, participants provided a rating of perceived breathlessness and their exertion using a simple 0−10 modified Borg scale (Borg, 1982), with zero being no difficulty and 10 being maximal difficulty.

## Data analysis

Raw signals underwent analogue-to-digital conversion at 1 kHz (Powerlab and LabChart v8; ADInstruments) and were stored for offline analysis. Electrocardiogram (BioAmp, FE231, ADInstruments, Bella Vista, NSW, Australia), systolic BP (SBP), diastolic BP (DBP) and mean BP (Finometer PRO, Finapres Medical Systems) (Bogert & van Lieshout, 2005), oxygen saturation ($S_pO_2$) (MLT321 and ML320/F, ADInstruments) and respiration (3830 Series, Heated Linear E Pneumotachometer, Hans Rudolph Inc., Kansas City, MO, USA; Respiratory Gas Analyzer, ML206, ADInstruments, Australia) were recorded simultaneously. Brachial artery diameter and blood velocity (Ogoh et al., 2009; Radegran, 1997) were recorded simultaneously using duplex Doppler ultrasound (uSmart 3300, Terason). Ultrasound images were screen-captured and stored as digital AVI files for offline analysis (Cardiovascular Suite, QUIPU, Italy) by a single operator (A.L.C.S.).

Beat-to-beat and breath-by-breath data were identified, and baseline was taken as the average over the last 3 min of rest. For isocapnic hypoxia rebreathing, the last 15 s were used as the 'peak hypoxia rebreathing' response. In addition, breath-by-breath $SpO_2$ from the isocapnic hypoxia rebreathing period was bin-averaged (3 mmHg bins), plotted against minute ventilation ($\dot{V}_E$) and slope fitted with a linear regression (least squares fit) using RStudio software (v2023.09.1, Posit Software, Boston, MA, USA). The peripheral chemoreflex sensitivity was calculated as the $\dot{V}_E$ *versus* $SpO_2$ slope.

Brachial artery monitoring was undertaken continuously for 5 min: 1 min of rest, 3 min of rhythmic handgrip exercise and 1 min of recovery. Brachial blood flow was calculated as mean blood velocity/2 × ($\pi$ × (diameter/2)$^2$) × 60, and vascular conductance was calculated as brachial blood flow / mean BP.

**Table 2. Cardiorespiratory and brachial arterial haemodynamic variables at baseline during saline and dopamine infusion**

|  | Baseline | | |
|---|---|---|---|
|  | Saline | Dopamine | *P* value |
| **Respiration** | | | |
| $SpO_2$ (%) | 98 ± 1 | 97 ± 1 | **0.019** |
| $P_{ET}O_2$ (mmHg) | 98 ± 7 | 95 ± 8 | **0.020** |
| $P_{ET}CO_2$ (mmHg) | 42 ± 3 | 43 ± 3 | **0.021** |
| $\dot{V}_E$ (l min$^{-1}$) | 11.5 ± 3.1 | 10.5 ± 3.2 | **0.036** |
| $V_T$ (l) | 0.91 ± 0.29 | 0.87 ± 0.26 | 0.189 |
| R*f* (breaths min$^{-1}$) | 14 ± 7 | 14 ± 8 | 0.253 |
| Perception of breathlessness (a.u.) | 0.71 ± 0.27 | 0.04 ± 0.14 | **<0.0001** |
| **Cardiovascular** | | | |
| HR (beats min$^{-1}$) | 64 ± 7 | 67 ± 7 | **0.008** |
| SBP (mmHg) | 136 ± 12 | 132 ± 9 | 0.137 |
| DBP (mmHg) | 80 ± 11 | 77 ± 11 | 0.105 |
| Mean BP (mmHg) | 98 ± 10 | 95 ± 11 | 0.155 |
| **Brachial haemodynamics** | | | |
| Diameter (mm) | 4.28 ± 1.30 | 4.37 ± 1.75 | 0.766 |
| Blood flow (ml min$^{-1}$) | 49.9 ± 17.1 | 56.1 ± 18.9 | **0.043** |
| Blood flow velocity (m s$^{-1}$) | 17.2 ± 5.5 | 18.0 ± 5.6 | 0.254 |
| Vascular conductance (ml min$^{-1}$ mmHg$^{-1}$) | 0.483 ± 0.160 | 0.565 ± 0.246 | **0.039** |

Cardiorespiratory variables are *n* = 14, and brachial haemodynamics are *n* = 12. Values are expressed as the mean ± SD. Abbreviations: $SpO_2$, oxygen saturation; $P_{ET}O_2$, partial pressure of end-tidal oxygen; $P_{ET}CO_2$, partial pressure of end-tidal carbon dioxide; $\dot{V}_E$, minute ventilation; $V_T$, tidal volume; R*f*, breathing frequency; HR, heart rate; SBP, systolic blood pressure; DBP, diastolic blood pressure, BP, blood pressure. *P* values represent paired Student's *t* test results. Bold font represents *P* < 0.05.

## Statistical analysis

Statistical analyses were performed using the statistical package IBM SPSS Statistics software (SPSS, version 29.0, IBM Corp., Armonk, NY, USA). Normality was assessed using the Shapiro–Wilk test. Paired Student's *t* test was used to compare cardiorespiratory variables between saline and dopamine trials. During rhythmic handgrip exercise, the main effects of drug (saline, dopamine), time (rest, handgrip) and their interaction were examined using mixed linear model analysis. *Post hoc* analysis was undertaken using *t* tests with Bonferroni correction. *P* < 0.05 was considered significant. Values are presented as means ± SD unless otherwise stated.

## Results

### Cardiorespiratory and brachial haemodynamics at baseline with saline and dopamine

Dopamine infusion resulted in a reduction in $SpO_2$ (*P* = 0.019), $P_{ET}O_2$ (*P* = 0.020), $\dot{V}_E$ (*P* = 0.036) and the perception of breathlessness (*P* < 0.0001) compared with saline at baseline (Table 2). Additionally, dopamine increased partial pressure of end-tidal carbon dioxide ($P_{ET}CO_2$) (*P* = 0.021), heart rate (HR) (*P* = 0.008), brachial blood flow (*P* = 0.043) and vascular conductance

(*P* = 0.039) compared with saline, while SBP, DBP and mean BP were unchanged (all *P* > 0.05).

### Cardiorespiratory responses to peripheral chemoreflex activation with saline and dopamine

Table 3 shows the cardiorespiratory responses to hypoxic rebreathing (last 15 s). Dopamine infusion blunted the peak increase in $\dot{V}_E$ (50 ± 43 *vs.* 68 ± 35%, *P* = 0.049), SBP (6 ± 8 *vs.* 13 ± 10%, *P* = 0.006), DBP (1 ± 4 *vs.* 5 ± 6%, *P* = 0.030) and mean BP (3 ± 5 *vs.* 8 ± 7%, *P* = 0.007) compared with saline. No differences between saline and dopamine were observed at peak hypoxic rebreathing for $SpO_2$, $P_{ET}CO_2$, tidal volume ($V_T$), breathing frequency (R*f*) and HR. A 28 ± 37% decrease in peripheral chemoreflex sensitivity was observed during hypoxic rebreathing with dopamine compared with saline (−0.477 ± 0.412 *vs.* −0.555 ± 0.464 l min$^{-1}$ %$^{-1}$, *P* = 0.044) infusion (Fig. 1).

### Cardiorespiratory and brachial haemodynamic responses to rhythmic handgrip exercise with saline and dopamine

During rhythmic handgrip exercise, force production at 1 min (55 ± 4 *vs.* 55 ± 4%), 2 min (55 ± 5 *vs.*

**Table 3. Cardiorespiratory response at peak hypoxic rebreathing during saline and dopamine infusion.**

| | Peak rebreathing | | |
| --- | --- | --- | --- |
| | Saline | Dopamine | *P* value |
| ***Respiration*** | | | |
| $\triangle SpO_2$ (%) | $-16 \pm 1$ | $-16 \pm 1$ | 0.480 |
| $\triangle P_{ET}O_2$ (mmHg) | $-52 \pm 8$ | $-49 \pm 8$ | **0.023** |
| $\triangle P_{ET}CO_2$ (mmHg) | $-2 \pm 2$ | $-1 \pm 2$ | 0.230 |
| $\triangle \dot{V}_E$ (l min$^{-1}$) | $8.9 \pm 5.7$ | $6.4 \pm 6.2$ | **0.026** |
| $\triangle V_T$ (l) | $0.55 \pm 0.32$ | $0.49 \pm 0.42$ | 0.243 |
| $\triangle Rf$ (breaths min$^{-1}$) | $0 \pm 3$ | $-1 \pm 3$ | 0.184 |
| $\triangle$Perception of breathlessness (a.u.) | $2 \pm 2$ | $2 \pm 2$ | 0.381 |
| ***Cardiovascular*** | | | |
| $\triangle HR$ (beats min$^{-1}$) | $9 \pm 4$ | $9 \pm 7$ | 0.298 |
| $\triangle SBP$ (mmHg) | $17 \pm 13$ | $8 \pm 11$ | **0.006** |
| $\triangle DBP$ (mmHg) | $4 \pm 5$ | $1 \pm 3$ | **0.036** |
| $\triangle$Mean BP (mmHg) | $8 \pm 7$ | $3 \pm 5$ | **0.007** |

Cardiorespiratory variables are *n* = 14. Values are expressed as the mean ± SD.
Abbreviations: SpO$_2$, oxygen saturation; P$_{ET}$O$_2$, partial pressure of end-tidal oxygen; P$_{ET}$CO$_2$, partial pressure of end-tidal carbon dioxide; $\dot{V}_E$, minute ventilation; $V_T$, tidal volume; R*f*, breathing frequency; HR, heart rate; SBP, systolic blood pressure; DBP, diastolic blood pressure, BP, blood pressure. *P* values represent paired Student's *t* test results. Bold font represents *P* < 0.05.

56 ± 4%) and 3 min (54 ± 5 *vs.* 55 ± 4%) were not different between saline and dopamine infusion (drug effect *P* = 0.475, time effect *P* = 0.688; drug × time effect *P* = 0.949). Brachial blood flow (*P* < 0.0001) and vascular conductance (*P* < 0.0001) increased from rest after 45 s of rhythmic handgrip exercise and then remained elevated throughout (Fig. 2). Importantly, brachial blood flow (*P* = 0.025) and vascular conductance (*P* = 0.021) responses to handgrip were greater with dopamine than saline. Mean BP (Fig. 2), HR and $\dot{V}_E$ (Fig. 3) were elevated from rest during rhythmic handgrip exercise (*P* > 0.05), but no differences were observed between dopamine and saline.

## Discussion

The aim of this study was to determine whether the administration of low-dose dopamine to inhibit the peripheral chemoreflex would improve the blood flow to the exercising muscles in humans with hypertension. The major novel findings are that (1) low-dose dopamine decreased resting $\dot{V}_E$, the hypoxic ventilatory response, and increased brachial blood flow and vascular conductance compared with saline at rest, and (2) rhythmic handgrip exercise-induced increases in brachial blood flow and vascular conductance were greater with low-dose dopamine than saline. Collectively, these findings are consistent with the view that the peripheral chemoreflex is tonically active at both rest and exercise, where it restrains blood flow and increases vascular conductance in human hypertension.

As recently reviewed (Felippe, Rio et al., 2023), carotid body hypertonicity and hyperreflexia are cardinal features of the spontaneously hypertensive rat (Abdala et al., 2012; McBryde et al., 2013; Pijacka et al., 2016), while in human hypertension more pronounced sympathetic

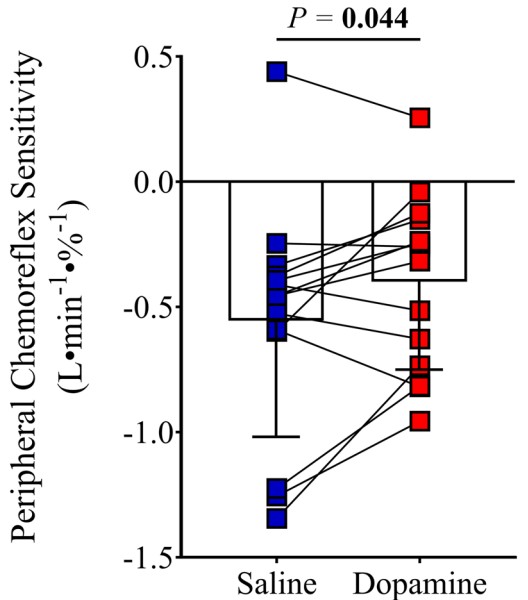

*P* = **0.044**

**Figure 1. Peripheral chemoreflex responsiveness with saline and dopamine**
Data are *n* = 14. Bars represent the mean for each trial, and symbols indicate the values of each participant.

responses to hypoxia and hyperoxia have been reported (Sinski et al., 2012; Somers et al., 1988). We observed that low-dose dopamine reduced baseline $\dot{V}_E$ and increased $P_{ET}CO_2$, indicative of ventilatory suppression and the tonic activation of the peripheral chemoreflex at rest in human hypertension. These findings are in contrast to previous work in healthy participants of a similar age in whom low-dose dopamine infusion failed to reduce resting $\dot{V}_E$ (Edgell et al., 2015). However, while we did not observe a change in BP at baseline with dopamine, brachial blood flow and vascular conductance were increased compared with saline. Such findings may be explained by the antagonism of the peripheral chemoreflex, resulting in sympatho-inhibition. Unfortunately, the challenging nature of the experimental protocol for the participants in the present study meant that we were unable to include a measure of muscle sympathetic nerve activity to verify this.

Evidence that the low-dose dopamine we used engaged with its target (i.e. peripheral chemoreceptors) includes the reduction of baseline $\dot{V}_E$ and the attenuated hypoxic ventilatory response. However, low-dose dopamine (i.e. $<5$ μg kg min$^{-1}$) may cause vasodilatation and increased blood flow via stimulation of peripheral dopaminergic receptors (Limberg et al., 2016), predominantly expressed in the renal and mesenteric regions

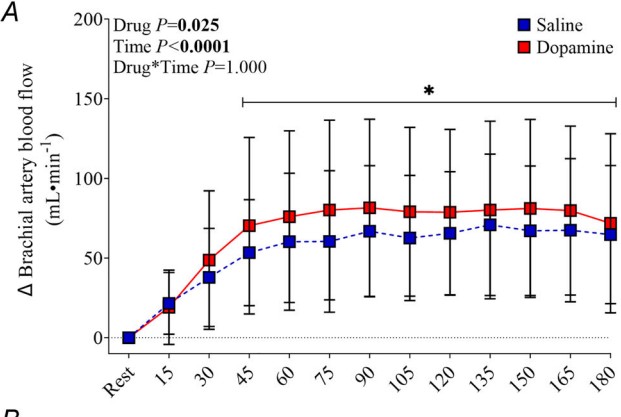

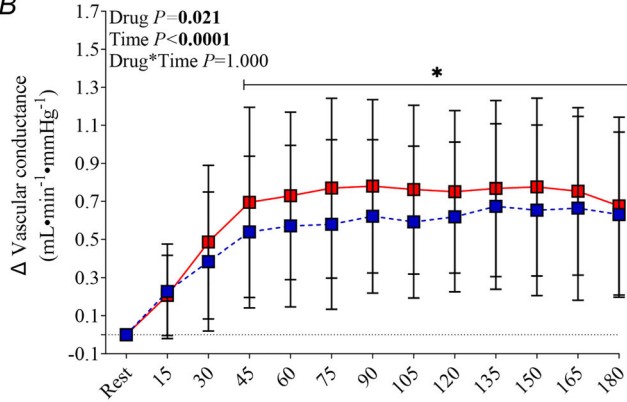

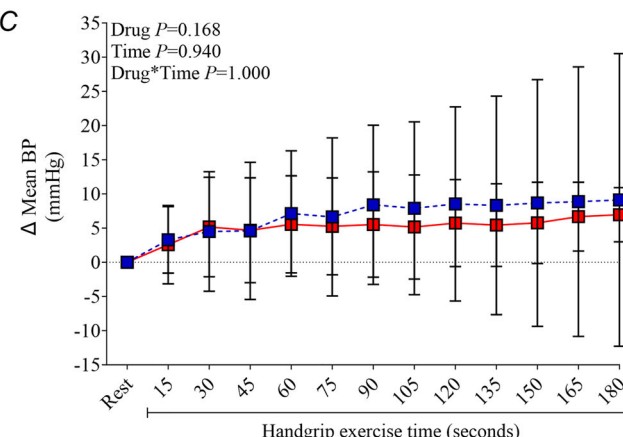

**Figure 2. Brachial artery blood flow (*A*), vascular conductance (*B*) and mean BP (*C*) responses to rhythmic handgrip exercise with saline and dopamine**
Brachial arterial haemodynamic variables are *n* = 12 and cardiorespiratory variables are *n* = 14. Values are expressed as the mean ± SD. BP, blood pressure. * Denotes *P* < 0.05 *versus* rest.

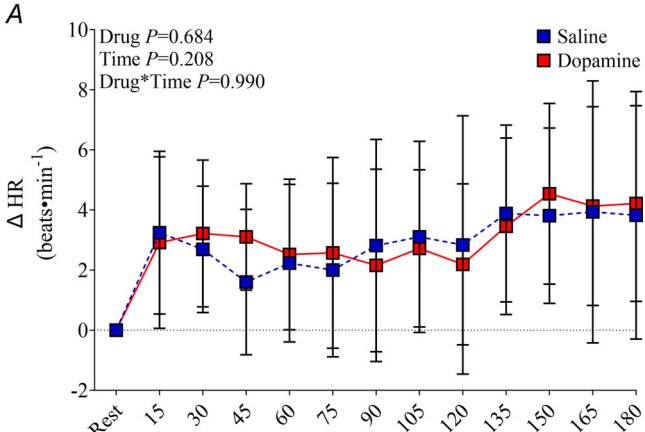

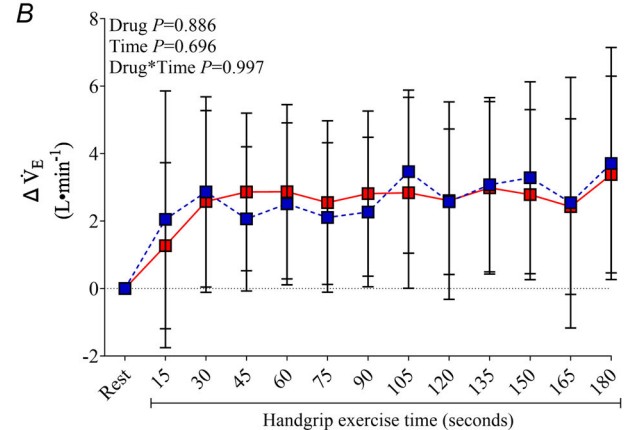

**Figure 3. Heart rate (*A*) and minute ventilation (*B*) responses to rhythmic handgrip exercise with saline and dopamine**
Cardiorespiratory variables are *n* = 14. Values are expressed as the mean ± SD. HR, heart rate; $V_E$, minute ventilation.

(Clark & Menninger, 1980). Indeed, targeting peripheral dopaminergic receptors with low-dose dopamine has been explored as a means of increasing renal blood flow in critically ill patients (Davis et al., 1982; Duke & Bersten, 1992). However, in healthy controls, low-dose dopamine has not been consistently observed to evoke vasodilatation (Bryan et al., 2012; Niewinski et al., 2014; Stickland et al., 2011). For example, Stickland et al. (2011) observed no change in baseline femoral blood flow and conductance in young healthy participants with the same dose of dopamine as the present study. Such differences may relate to the varied circulations studied (e.g. arm, leg) and individual differences in low-dopamine sensitivity (Limberg et al., 2016) and/or peripheral chemoreflex sensitivities. Of note, Edgell et al. (2015) observed that medically managed heart failure patients exhibited both decreased ventilation and total peripheral resistance index at rest in response to dopamine, which is relevant because heightened peripheral chemoreflex sensitivity is reported in heart failure (Ponikowski et al., 2001). Our findings, and those of Edgell et al. (2015), highlight the fact that despite medical pharmacotherapy, the peripheral chemoreflex can still exert important cardiorespiratory control in clinical conditions. While beyond the scope of the current study, future investigations should explore whether the response to peripheral chemoreflex inhibition is related to the anti-hypertensive medications being used.

Increases in sympathetic vasoconstrictor activity during exercise are traditionally attributed to the actions of central command and particularly the exercise pressor reflex (Fisher et al., 2015), but more recently, a role for the peripheral chemoreflex has also emerged (Stickland et al., 2008). Known to cause sympatho-excitation in response to hypoxia, the peripheral chemoreceptors are also activated by a host of stimuli (e.g. heat, $H^+$, $K^+$, glucose, sympatho-excitation (Felippe, Zera et al., 2023; Zera et al., 2019)), which all increase during exercise. Furthermore, peripheral chemoreflex sensitivity increases from rest to exercise (Oliveira et al., 2024) and its pharmacological and physiological antagonism has been shown to have notable effects on neural circulatory control in animal and human studies. For instance, Stickland and colleagues (Stickland et al., 2007) observed that close-carotid injection of dopamine caused a rapid hindlimb blood flow increase at rest and during exercise in canines with heart failure. In humans, transient peripheral chemoreflex inhibition with 100% inspired oxygen decreased muscle sympathetic nerve activity during handgrip exercise (Stickland et al., 2008), and both transient hyperoxia and low-dose dopamine increased femoral artery blood flow and vascular conductance during knee-extension exercise (Stickland et al., 2011). This is in line with the findings of the present study that low-dose dopamine augments the brachial blood flow and vascular conductance response to rhythmic handgrip exercise in human hypertension. Collectively, such findings suggest that, in disease states characterised by heightened peripheral chemoreflex tonicity/sensitivity, the infusion of low-dose dopamine to inhibit the peripheral chemoreceptors reduces sympathetic vasoconstrictor outflow and augments exercising skeletal muscle blood flow.

Animal and human studies have reported that the exercise pressor reflex is overactive in hypertension (Barbosa et al., 2016; Delaney et al., 2010; Leal et al., 2013; Sausen et al., 2009), although this is not a universal finding (Rondon et al., 2006). Nevertheless, it is tempting to speculate that a raised sensitivity of group III and IV skeletal muscle afferents in hypertension provides a mechanism whereby peripheral chemoreflex sensitivity is robustly raised during exercise (Oliveira et al., 2024), perhaps via enhancing sympathetic activity to the carotid bodies (Felippe, Zera et al., 2023), and contributes to our observation that pharmacological inhibition of the peripheral chemoreceptors augments the hyperaemic response of the exercising skeletal muscle. Future studies should consider the interactive mechanisms by which the exercise pressor reflex and the peripheral chemoreflex may affect the hyperaemic response of the exercising skeletal muscle blood flow in human hypertension.

A rest, low-dose dopamine was associated with a reduction in ratings of breathlessness, that is, from $0.71 \pm 0.27$ to $0.04 \pm 0.14$ units (0–10 modified Borg scale (Borg, 1982), with zero being no difficulty and 10 being maximal difficulty). However, as the minimal clinically important difference for this scale has been reported as one unit (Oxberry et al., 2012), it is unlikely that the changes we observed are clinically significant. Although slight, the perception of breathlessness under the saline condition may relate to an elevated peripheral chemoreflex sensitivity in hypertensive participants or the breathing apparatus used, which, as is standard, included a bacterial filter and pneumotachometer that may have offered some resistance to eupnoeic breathing. Nevertheless, its attenuation with low-dose dopamine is in line with previous work implicating the peripheral chemoreflex as a potential contributor to the sensation of dyspnoea (Buchanan & Richerson, 2009). Notably, during rhythmic handgrip exercise, the ratings of breathlessness were increased (to $2 \pm 2$ units), but not different between the saline and low-dose dopamine trials.

There are several experimental factors that should be considered when evaluating the findings of the present study. First, no healthy control group was recruited. This is because the *a priori* study objective was to determine whether the infusion of low-dose dopamine to inhibit the peripheral chemoreflex will augment blood flow to exercising skeletal muscle in participants with

hypertension. In this study, we were not concerned with whether the magnitude of the responses observed in hypertension were greater than control participants. Second, we used a pharmacological approach (i.e. intravenous low-dose dopamine) to inhibit the peripheral chemoreflex. Previous work in exercising canines, more specifically, inhibited the carotid chemoreceptors by using a close-carotid injection of dopamine (Stickland et al., 2007), but this approach is not feasible in humans. We observed that low-dose dopamine reduced resting ventilation and attenuated the hypoxic ventilatory response, providing evidence for peripheral chemoreflex inhibition and being broadly consistent with previous reports (Bascom et al., 1991; Henson et al., 1992; Stickland et al., 2011; Welsh et al., 1978). However, it is a limitation of the present study that we did not determine the optimal dopamine dose to attenuate the hypoxic ventilatory response for each participant, as per the recommendations of Limberg et al. (2016), and as a consequence, peak peripheral chemoreflex inhibition may not have been achieved. As acknowledged above, there is potential for low-dose dopamine to cause vasodilatation via peripherally expressed dopaminergic receptors, but this effect is more associated with the renal and mesenteric circulations rather than that of the skeletal muscle, which has not been consistently observed to exhibit vasodilatation in healthy participants with low peripheral chemoreflex sensitivity (Bryan et al., 2012; Niewinski et al., 2014; Stickland et al., 2011). More importantly, the intravenous infusion of dopamine was constant throughout the baseline and rhythmic exercise periods, and as such, a local effect of dopamine is unlikely to explain the augmented hyperaemic response to rhythmic handgrip exercise that was noted (i.e. a greater change from baseline with dopamine). Therefore, we believe that these findings are consistent with the suppression of the peripheral chemoreflex in the regulation of regional blood flow during exercise. Secondary to its effects on the carotid chemoreceptors, dopamine may have caused a compensatory increase in aortic chemoreceptor responsiveness or other regulatory pathways. An alternative method of inhibiting the peripheral chemoreflex in humans would have been to administer hyperoxia (Chua et al., 1996; Edgell et al., 2015; Stickland et al., 2007, 2008). However, this is potentially confounded by the augmented systemic oxygen delivery and potential for altered skeletal muscle metabolism (e.g. attenuated acidosis). Finally, treated hypertension participants were studied, and it is possible that their prescribed medications independently affected peripheral chemoreflex sensitivity and exercise responses (Beloka et al., 2008; Leung et al., 2000). However, as the therapy received was standard, this broadens the relevance of our observations to treated hypertension.

## Perspectives

Heightened peripheral chemoreflex sensitivity is associated with poor outcomes (Ponikowski et al., 2001) and may drive sympatho-excitation and end-organ dysfunction in human hypertension (Felippe, Rio et al., 2023). Exercise-induced activation of the sympathetic nervous system is important for raising cardiac output, the appropriate distribution of cardiac output to the active skeletal muscle and maintenance of an appropriate blood pressure. However, if this sympatho-excitation is excessive and causes an excessive vasoconstrictor response that deprives the active skeletal muscles of nutritive blood flow, premature fatigue, exercise intolerance and exaggerated sympathetic responses with surges in blood pressure can occur (Joyner & Casey, 2015), even in treated hypertension (Chant et al., 2018). This could increase cardiovascular risk despite regular exercise being advised for lowering BP in hypertension (Pescatello et al., 2015). Therefore, our observation that suppression of peripheral chemoreflex sensitivity improves active skeletal muscle blood flow in participants with treated hypertension implicates peripheral chemoreceptors as a potential target for improving exercise safety, tolerability and adherence, which may help hypertensive participants access the benefits of regular exercise participation.

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

## Additional information

### Data availability statement

Data are available from the corresponding author upon reasonable request.

## Competing interests

The authors have no conflicts of interest to declare.

## Author contributions

The experiments were performed in the Human Physiology Laboratory at the University of Auckland, New Zealand. A.L.C.S.: acquisition, analysis or interpretation of data for the work; drafting the work or revising it critically for important intellectual content. M.J.P.: acquisition, analysis or interpretation of data for the work; drafting the work or revising it critically for important intellectual content. T.B.: acquisition, analysis or interpretation of data for the work; drafting the work or revising it critically for important intellectual content. M.D.: conception or design of the work; drafting the work or revising it critically for important intellectual content. J.F.R.P.: conception or design of the work; acquisition, analysis or interpretation of data for the work; drafting the work or revising it critically for important intellectual content. J.P.F.: conception or design of the work; acquisition, analysis or interpretation of data for the work; drafting the work or revising it critically for important intellectual content. All authors approved the final version of the manuscript, agree to be accountable for all aspects of the work in ensuring that questions related to the accuracy or integrity of any part of the work are appropriately investigated and resolved, and all persons designated as authors qualify for authorship, and all those who qualify for authorship are listed.

## Funding

Support was provided by the Health Research Council of New Zealand (Ref. no. 19/687. J.P.F., J.F.R.P.), the Greenlane Research and Educational Fund (Ref# 21/01/4153; M.J.P.), and the Sydney Taylor Trust (J.F.R.P.).

## Acknowledgements

The authors wish to thank all the volunteers for their enthusiastic participation in this study.

Open access publishing facilitated by The University of Auckland, as part of the Wiley - The University of Auckland agreement via the Council of Australian University Librarians.

## Keywords

chemoreceptor-reflex, hypertension, sympathetic nerve activity

## Supporting information

Additional supporting information can be found online in the Supporting Information section at the end of the HTML view of the article. Supporting information files available:

**Peer Review History**

