## [Peer Review History · The Journal of Physiology]

Peripheral chemoreflex restrains skeletal muscle blood flow during exercise in participants with treated hypertension

Ana Luiza C Sayegh, Michael J Plunkett, Thalia Babbage, Mathew Dawes, Julian F. R. Paton, and James P Fisher
DOI: 10.1113/JP286998

Corresponding author(s): James Fisher (jp.fisher@auckland.ac.nz)

The following individual(s) involved in review of this submission have agreed to reveal their identity: Michael K Stickland (Referee #1)

Review Timeline:

Submission Date:	26-May-2024
Editorial Decision:	25-Jun-2024
Revision Received:	15-Jul-2024
Accepted:	01-Aug-2024

Senior Editor: Vaughan Macefield

Reviewing Editor: Marc Kaufman

Transaction Report:

Dear Dr Fisher,

Re: JP-RP-2024-286998 "Peripheral chemoreflex restrains skeletal muscle blood flow during exercise in participants with treated hypertension" by Ana Luiza C Sayegh, Michael J Plunkett, Thalia Babbage, Mathew Dawes, Julian F. R. Paton, and James P Fisher

Thank you for submitting your manuscript to The Journal of Physiology. It has been assessed by a Reviewing Editor and by 2 expert referees and we are pleased to tell you that it is acceptable for publication following satisfactory revision.

REVISION CHECKLIST:

- 'Potential Cover Art' for consideration as the issue's cover image

- Appropriate Supporting Information (Video, audio or data set: see https://jp.msubmit.net/cgi-bin/main.plex?form_type=display_requirements#supp).

We look forward to receiving your revised submission.

Yours sincerely,

Vaughan Macefield
Senior Editor
The Journal of Physiology

REQUIRED ITEMS

- Author photo and profile. First or joint first authors are asked to provide a short biography (no more than 100 words for one author or 150 words in total for joint first authors) and a portrait photograph. These should be uploaded and clearly labelled together in a Word document with the revised version of the manuscript. See Information for Authors for further details.

- Please upload separate high-quality figure files via the submission form.

- Please ensure that any tables are editable and in Word format, and wherever possible, embedded in the article file itself.

- Please ensure that the Article File you upload is a Word file.

- Please include an Abstract Figure file, as well as the Figure Legend text within the main article file. The Abstract Figure is a piece of artwork designed to give readers an immediate understanding of the research and should summarise the main conclusions. If possible, the image should be easily 'readable' from left to right or top to bottom. It should show the physiological relevance of the manuscript so readers can assess the importance and content of its findings. Abstract Figures should not merely recapitulate other figures in the manuscript. Please try to keep the diagram as simple as possible and without superfluous information that may distract from the main conclusion(s). Abstract Figures must be provided by authors no later than the revised manuscript stage and should be uploaded as a separate file during online submission labelled as File Type 'Abstract Figure'. Please also ensure that you include the figure legend in the main article file. All Abstract Figures should be created using BioRender. Authors should use The Journal's premium BioRender account to export high-resolution images. Details on how to use and access the premium account are included as part of this email.

EDITOR COMMENTS

Reviewing Editor:

Both reviewers thought that your manuscript made an important contribution to the neural control of blood flow to resting and exercising legs in hypertensive patients. Their comments were minor and can be easily addressed. Please respond to them.

Please also see 'Required Items' above.

Senior Editor:

Dear James,

Thank you for resubmitting your manuscript to the special Call for Papers in The Journal of Physiology. I have now received comments from two independent assessors and a Reviewing Editor, all experts in the field. As you will see, there are a few

minor issues that will need addressing and I look forward to receiving your revised submission in due course.

All the best,

Vaughan Macefield

Senior Editor

REFEREE COMMENTS

Referee #1:

This study examined the influence of the carotid chemoreceptor on blood flow control during handgrip exercise in participants with treated hypertension. Using a randomized double-blinded controlled trial, IV dopamine was shown to increase resting and exercise blood flow and conductance as compared to placebo. This well-executed study clearly establishes that the carotid chemoreceptor plays an important role in blood flow regulation at rest and during exercise in people with hypertension.

Work by Joyner et al has shown that individual chemoreceptor inhibition may be optimized by establishing an individual's ideal IV dopamine dose (Limberg et al. 2016). The authors do a good job of providing evidence of chemoreceptor inhibition with dopamine, but the response may have been larger should they have determined each participant's optimal dopamine dose. The present investigators successfully completed some complicated experiments, but this limitation should be acknowledged.

A strength of the work is that the investigators examined patients who were already treated for their hypertension. This is important, as some of the early chemoreceptor heart failure work examined patients who were generally not well managed, and not on newer medications which we now know would affect the carotid chemoreceptor (e.g. ARBs). The authors could consider highlighting that despite pharmacotherapy, the CC still exerts important cardiovascular control, and suggests there is still room for better medical management. The authors would be encouraged to expand this work to see if the response to CC inhibition is at all related to underlying anti-hypertensive medications.

The investigators recruited treated hypertensive patients, and the medications are listed in Table 1. Presumably all patients were treated with at least one anti-hypertensive medication, but this is not totally clear by the participant description paragraph.

Referee #2:

Sayegh et al tested the hypothesis that increased sensitivity of the peripheral chemoreflex causes a sympathetic restraint of blood flow to exercising muscles. They tested 14 patients with treated hypertension under control (IV saline) conditions and during an IV infusion of dopamine to inhibit the peripheral chemoreflex, during baseline, isocapnic hypoxic rebreathing, and during rhythmic handgrip exercise. They found that low dose dopamine increased brachial blood flow and vascular conductance during baseline and exercise. Overall, they concluded that peripheral chemoreflex overactivity restrains blood flow during exercise in hypertensive humans.

1. Line 77: assume "these" refers to the exercise-induced surges in BP? This sentence will be clearer if rewritten.

2. Line 127: Consider adding the age range. Also were the participants instructed to continue medication usage as usual or

was there a withdrawal period from medications? For example, did they refrain from taking their medications on the morning of the study?

3. Lines 151-152: Although a reference is provided for the isocapnic hypoxia rebreathing protocol, consider adding a very brief description of the protocol.

4. Line 153: Was force production measured and is there any data to demonstrate participant adherence to the protocol between trials?

5. What was the time gap between the two trials? Is this sufficient for the washout of dopamine?

6. Lines 233-234: Specify the population studied in the study by Edgell et al (2015).

7. Lines 291-301: Although the rating of breathlessness decreased from 0.71 to 0.04 with dopamine on average, given this is on a 0-10 scale, this seems like a small change. Is this a clinically meaningful change?

8. Table 1: Delete "and normotensive" from the table title.

9. Table 3: What are the numbers in parenthesis for the CV parameters?

10. Figure 1: Consider connecting/pairing the data points since they are the same subjects - this will show how many of the subjects individually responded with a decline in sensitivity during the dopamine infusion.

END OF COMMENTS

Confidential Review

26-May-2024

We are grateful to the Reviewing Editor and both Referees for their careful consideration of our manuscript and the opportunity to submit a revised version. We believe that the insightful points raised have helped us improve our work's clarity and impact. As requested, we have provided a point-by-point response to the concerns raised (below) and both a clean version of our revised manuscript and a marked-up version showing the changes made in red underlined font (attached).

Reviewing Editor:

“Both reviewers thought that your manuscript made an important contribution to the neural control of blood flow to resting and exercising legs in hypertensive patients. Their comments were minor and can be easily addressed. Please respond to them.”

Response: Thank you for your support. We are grateful to the reviewers for their comments and have positively responded to all the points raised.

Senior Editor:

“Thank you for resubmitting your manuscript to the special Call for Papers in The Journal of Physiology. I have now received comments from two independent assessors and a Reviewing Editor, all experts in the field. As you will see, there are a few minor issues that will need addressing and I look forward to receiving your revised submission in due course.”

Response: Dear Prof Macefield thank you for your encouraging comments.

Referee #1:

1) *“...This well-executed study clearly establishes that the carotid chemoreceptor plays an important role in blood flow regulation at rest and during exercise in people with hypertension. Work by Joyner et al has shown that individual chemoreceptor inhibition may be optimised by establishing an individuals ideal IV dopamine dose (Limberg et al. 2016).*

The authors do a good job of providing evidence of chemoreceptor inhibition with dopamine, but the response may have been larger should they have determined each participant's optimal dopamine dose. The present investigators successfully completed some complicated experiments, but this limitation should be acknowledged.”

Response: Thank you for your comments. We now acknowledge that we did not determine an ideal iv dopamine dose for each participant according to the recommendations of Limberg et al., (2016) as a limitation of the present study (page 16, line 336).

2) *“A strength of the work is that the investigators examined patients who were already treated for their hypertension. This is important, as some of the early chemoreceptor heart failure work examined patients who were generally not well managed, and not on newer medications which we now know would affect the carotid chemoreceptor (e.g. ARBs). The authors could consider highlighting that despite pharmacotherapy, the CC still exerts important cardiovascular control, and suggests there is still room for better medical management. The authors would be encouraged to expand this work to see if the response to CC inhibition is at all related to underlying anti-hypertensive medications.”*

Response: Thank you for raising this important point. In the revised Discussion section, we now highlight the fact that despite our participants being treated for hypertension the peripheral (carotid) chemoreflex still exerts important cardiovascular control (page 13, line 274). We agree that expansion of this work to consider if peripheral (carotid) chemoreflex inhibition is related to underlying anti-hypertensive medications is important. However, this will need to be addressed in future studies because in the current study all participants were medicated and therefore additional sub-group analysis is not possible.

3) *“The investigators recruited treated hypertensive patients, and the medications are listed in Table 1. Presumably all patients were treated with at least one anti-hypertensive medication, but this is not totally clear by the participant description paragraph.”*

Response: Yes. All patients were treated with at least one anti-hypertensive medication. This information is now added to the participant description paragraph (page 6, line 129).

Referee #2:

“1. Line 77: assume "these" refers to the exercise-induced surges in BP? This sentence will be clearer if rewritten.”

Response: Yes, "these" refers to the exercise-induced surges in BP. The sentence has been rewritten so that this information is clearer (page 4, line 77).

“2. Line 127: Consider adding the age range. Also were the participants instructed to continue medication usage as usual or was there a withdrawal period from medications? For example, did they refrain from taking their medications on the morning of the study?”

Response: As requested, the participants' age range has been added (page 6, line 126). Participants were asked to refrain from taking their medications on the morning of the study (page 7, line 142).

“3. Lines 151-152: Although a reference is provided for the isocapnic hypoxia rebreathing protocol, consider adding a very brief description of the protocol.”

Response: A brief description of isocapnic hypoxia rebreathing protocol has been added to revised Experimental Protocol section (page 7, line 160).

“4. Line 153: Was force production measured and is there any data to demonstrate participant adherence to the protocol between trials?”

Response: Force production was measured during rhythmic handgrip exercise with saline and dopamine. There were no differences in force production during rhythmic handgrip exercise for drug (saline vs dopamine, $P=0.475$), time (minute 1, minute 2 or minute 3, $P=0.688$) and the drug x time interaction ($P=0.949$). This information has been added to the revised Result section (page 10, line 224).

“5. What was the time gap between the two trials? Is this sufficient for the washout of dopamine?”

Response: In this protocol, a 30-minute wash-out period was taken following the drug infusion (either saline or dopamine), which is well in excess of the half-life for dopamine in the systemic circulation which is 1 to 5 minutes (Sonne et al., 2024). This information has been added to the revised Manuscript (page 7, line 152).

“6. Lines 233-234: Specify the population studied in the study by Edgell et al (2015).”

Response: That Edgell et al (2015) studied healthy participants has been specified (page 12, line 250).

“7. Lines 291-301: Although the rating of breathlessness decreased from 0.71 to 0.04 with dopamine on average, given this is on a 0-10 scale, this seems like a small change. Is this a clinically meaningful change?”

Response: The minimal clinically important difference for the rating of perceived breathlessness (0-10 modified Borg scale (Borg, 1982)) is 1 unit for patients with chronic heart failure (Oxberry et al., 2012). We now acknowledge that the reduction we observed in

the rating of breathlessness (i.e., from 0.71 to 0.04) is below this threshold in the revised Discussion section (page 15, line 313).

“8. Table 1: Delete "and normotensive" from the table title.”

Response: Done.

“9. Table 3: What are the numbers in parenthesis for the CV parameters?”

Response: The numbers in parenthesis were a typographical error and have been deleted in the revised Table 3.

“10. Figure 1: Consider connecting/pairing the data points since they are the same subjects - this will show how many of the subjects individually responded with a decline in sensitivity during the dopamine infusion.”

Response: Done.

Reference:

Sonne J, Goyal A & Lopez-Ojeda W. (2024). Dopamine. In StatPearls. Treasure Island (FL).

Borg GA. (1982). Psychophysical bases of perceived exertion. *Med Sci Sports Exerc* 14, 377-381.

Oxberry SG, Bland JM, Clark AL, Cleland JG & Johnson MJ. (2012). Minimally clinically important difference in chronic breathlessness: every little helps. *Am Heart J* **164**, 229-235.

Dear Dr Fisher,

Re: JP-RP-2024-286998R1 "Peripheral chemoreflex restrains skeletal muscle blood flow during exercise in participants with treated hypertension" by Ana Luiza C Sayegh, Michael J Plunkett, Thalia Babbage, Mathew Dawes, Julian F. R. Paton, and James P Fisher

We are pleased to tell you that your paper has been accepted for publication in The Journal of Physiology.

Authors should note that it is too late at this point to offer corrections prior to proofing. Major corrections at proof stage, such as changes to figures, will be referred to the Editors for approval before they can be incorporated. Only minor changes, such as to style and consistency, should be made at proof stage. Changes that need to be made after proof stage will usually require a formal correction notice.

If you would like to receive our 'Research Roundup', a monthly newsletter highlighting the cutting-edge research published in The Physiological Society's family of journals (The Journal of Physiology, Experimental Physiology and Physiological Reports), please click this link, fill in your name and email address and select 'Research Roundup': <https://www.physoc.org/journals-and-media/membernews/>.

Yours sincerely,

Vaughan Macefield
Senior Editor
The Journal of Physiology

P.S. - You can help your research get the attention it deserves! Check out Wiley's free Promotion Guide for best-practice recommendations for promoting your work at www.wileyauthors.com/eeo/guide. You can learn more about Wiley Editing Services which offers professional video, design, and writing services to create shareable video abstracts, infographics, conference posters, lay summaries, and research news stories for your research at www.wileyauthors.com/eeo/promotion.

IMPORTANT NOTICE ABOUT OPEN ACCESS: To assist authors whose funding agencies mandate public access to published research findings sooner than 12 months after publication, The Journal of Physiology allows authors to pay an Open Access (OA) fee to have their papers made freely available immediately on publication.

You can check if your funder or institution has a Wiley Open Access Account here: <https://authorservices.wiley.com/author-resources/Journal-Authors/licensing-and-open-access/open-access/author-compliance-tool.html>.

EDITOR COMMENTS

Reviewing Editor:

Thank you for being responsive to the reviewers' critiques.

REFEREE COMMENTS

Referee #1:

The authors have addressed all my concerns. Congratulations on a great article.

Referee #2:

No additional comments.

1st Confidential Review

15-Jul-2024